# A Bayesian approach to infer recombination patterns in coronaviruses

Nicola F. Müller [1 ✉], Kathryn E. Kistler[1,2] & Trevor Bedford[1,2,3]

As shown during the SARS-CoV-2 pandemic, phylogenetic and phylodynamic methods are essential tools to study the spread and evolution of pathogens. One of the central assumptions of these methods is that the shared history of pathogens isolated from different hosts can be described by a branching phylogenetic tree. Recombination breaks this assumption. This makes it problematic to apply phylogenetic methods to study recombining pathogens, including, for example, coronaviruses. Here, we introduce a Markov chain Monte Carlo approach that allows inference of recombination networks from genetic sequence data under a template switching model of recombination. Using this method, we first show that recombination is extremely common in the evolutionary history of SARS-like coronaviruses. We then show how recombination rates across the genome of the human seasonal coronaviruses 229E, OC43 and NL63 vary with rates of adaptation. This suggests that recombination could be beneficial to fitness of human seasonal coronaviruses. Additionally, this work sets the stage for Bayesian phylogenetic tracking of the spread and evolution of SARS-CoV-2 in the future, even as recombinant viruses become prevalent.

[1] Vaccine and Infectious Disease Division, Fred Hutchinson Cancer Research Center, Seattle, WA, USA. [2] Molecular and Cellular Biology Program, University of Washington, Seattle, WA, USA. [3] Howard Hughes Medical Institute, Seattle, WA, USA. ✉email: nicola.felix.mueller@gmail.com

Since the emergence of SARS-CoV-2, genetic sequence data has been used to study its evolution and spread. Genetic sequences have, for example, been used to investigate natural versus lab origins of SARS-CoV-2[1], when SARS-CoV-2 was introduced into the US[2] as well as whether genetic variants differ in growth rate[3]. These analyses often rely on phylogenetic and phylodynamic approaches, at the heart of which are phylogenetic trees. Such trees denote how viruses isolated from different individuals are related and contain information about the transmission dynamics connecting these infections[4].

Along with mutations introduced by errors during replication or by anti-viral molecules (for example ref. [5]), different recombination processes contribute to genetic diversity in RNA viruses (reviewed by Simon-Loriere and Holmes[6]). Reassortment in segmented viruses (generally negative-sense RNA viruses), such as influenza or rotaviruses, can produce offspring that carry segments from different parental lineages[7]. In other RNA viruses (generally positive-sense RNA viruses), such as flaviviruses and coronaviruses, homologous recombination can combine different parts of a genome from different parental lineages in absence of physically separate segments on the genome of those viruses[8]. The main mechanism of this process is thought to be via template switching[9], where the template for replication is switched during the replication process. Recombination breakpoints in experiments appear to be largely random, with selection selecting recombination breakpoints in some areas of the genome[10]. Recent work shows that recombination breakpoints occur more frequently in the spike region of betacoronaviruses, such as SARS-CoV-2[11]. While the reason why the recombination process evolved in RNA viruses is not completely understood[6], there are different explanations of why recombination may be beneficial. In general, recombination is selected if breaking up the linkage disequilibrium is beneficial[12]. Recombination can help purge deleterious mutations from the genome, such as proposed by the mutational-deterministic hypothesis[13]. It can also increase the rate at which a fit combination of mutations occurs, such as stated by the Robertson–Hill effect[14]. Alternatively, recombination in RNA viruses may also just be a by-product of the processivity of the viral polymerase[6].

Recombination poses a unique challenge to phylogenetic methods, as it violates the very central assumption that the evolutionary history of individuals can be denoted by branching phylogenetic trees. Recombination breaks this assumption and requires representation of the shared ancestry of a set of sequences as a network. Not accounting for this can lead to biased phylogenetic and phylodynamic inferences[15,16]. An analytic description of recombination is provided by the coalescent with recombination, which models a backward in time process where lineages can coalesce and recombine[17]. When recombination is considered backward in time, a single lineage results in two-parent lineages, with one parent lineage carrying the genetic material from one side of a random recombination breakpoint and the other parent lineage carrying the genetic material from the other side of this breakpoint. This equates to the backward in time equivalent of template switching where there is one recombination breakpoint per recombination event.

Currently, some Bayesian phylogenetic approaches exist that infer recombination networks, or ancestral recombination graphs (ARG), but are either approximate or do not directly allow for efficient model-based inference. Some approaches consider tree-based networks[18,19], where the networks consist of a base tree with recombination edges that always attach to edges on the base tree. Alternative approaches rely on approximations to the coalescent with recombination[20,21], consider a different model of recombination[16], or seek to infer recombination networks absent an explicit recombination model[22]. Bayesian and maximum

likelihood methods have also been used to account for gene transfer events when describing the evolutionary history of species from multiple loci (for example, see refs. [23,24]). Additionally, methods have been used to describe non-tree-like evolution using split trees[25,26]. There is, however, a gap for Bayesian inference of recombination networks under the coalescent with recombination that can be applied to study pathogens, such as coronaviruses.

In order to fill this gap, we here develop a Markov chain Monte Carlo (MCMC) approach to efficiently infer recombination networks under the coalescent with recombination for sequences sampled over time. This framework allows joint estimation of recombination networks, effective population sizes, recombination rates, and parameters describing mutations over time from genetic sequence data sampled through time. We explicitly do not make additional approximations to characterize the recombination process, other than those of the coalescent with recombination[17], such as, for example, the approximation of tree-based networks. We implemented this approach as an open-source software package for BEAST2[27], allowing us to use the various evolutionary models already implemented in BEAST2. We then use the coalescent with recombination to study the recombination patterns of SARS-like, MERS, and 3 seasonal human coronaviruses.

## Results

**Widespread recombination in SARS-like coronaviruses**. Recombination has been implicated at the beginning of the SARS-CoV-1 outbreak[28] and has been suggested as the origin of the receptor-binding domain in SARS-CoV-2[29], though Boni et al.[30] report that recombination is unlikely to be the origin of SARS-CoV-2. While this strongly suggests non-tree-like evolution, the evolutionary history of SARS-like viruses has, out of necessity, mainly been denoted using phylogenetic trees.

We here reconstruct the recombination history of SARS-like viruses, which includes SARS-CoV-1 and SARS-CoV-2 as well as related bat[31–33] and pangolin[34] coronaviruses. To do so, we infer the recombination network of SARS-like viruses under the coalescent with recombination. We assumed that the rates of recombination and effective population sizes were constant over time and that the genomes evolved under a GTR+$\Gamma_4$ model. Similar to the estimate in ref. [30], we used a fixed evolutionary rate of $5 \times 10^{-4}$ mutations per nucleotide and year. We fixed the evolutionary rate since the time interval of sampling between individual isolates is relatively short compared to the time scale of the evolutionary history of SARS-like viruses. This means that the sampling times themselves offer little insight into the evolutionary rates and, in absence of other calibration points, there is little information about the evolutionary rate in this dataset. This, in turn, means that if the evolutionary rate we used here is inaccurate then the timings of common ancestors will also be inaccurate. Therefore, exact timings and calendar dates in this analysis should be taken as guideposts rather than formal estimates.

As shown in Fig. 1A and Fig. S1A, the evolutionary history of SARS-like viruses are characterized by frequent recombination events, including ancestral to SARS-CoV-2 (see Fig. S2). This means that only relatively short segments of the genomes code for the same tree (see Figs. S3 and S1B). Consequently, characterizing the evolutionary history of SARS-like viruses by a single, genome-wide phylogeny is bound to be inaccurate and potentially misleading. We infer the recombination rate in SARS-like viruses to be approximately $2 \times 10^{-6}$ recombination events per site per year, which is about 200 times slower than the evolutionary rate. This rate translated to about 0.06 recombination events per lineage per year, which is slightly lower than the estimated rate of

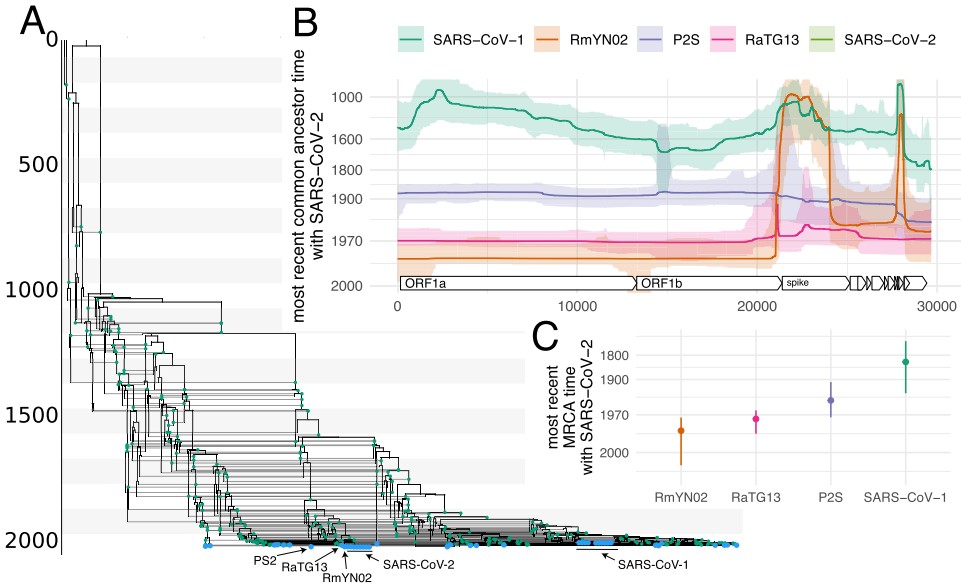

**Fig. 1 Evolutionary history of SARS-like viruses. A** Maximum clade credibility network of SARS-like viruses. Blue dots denote samples and green dots recombination events. **B** Common ancestor times of Wuhan-Hu1 (SARS-CoV-2) with different SARS-like viruses on different positions of the genome. The *y*-axis denotes common ancestor times on the log scale. The line denotes the median common ancestor time, while the colored area denotes the 95% highest posterior density interval. **C** Most recent time anywhere on the genome that Wuhan-Hu1 shared a common ancestor with different SARS-like viruses. The error bars denote the upper and lower bound of the 95% highest posterior density interval. The MCC network and common ancestor times are provided as a Source Data file.

recombination for the seasonal human coronaviruses and the reassortment rates for pandemic 1918 like influenza A/H1N1 and influenza B viruses, which are all around $0.1-0.2$ reassortment events per lineage per year[16]. This recombination rate is a function of co-infection rates, probability of recombination occurring upon co-infection, and selection. As such, the recombination rate we infer here will be (possibly substantially) lower than the within-host rate of recombination.

These recombination events were not evenly distributed across the genome and, instead, were relatively higher in areas outside those coding for ORF1ab (Figs. S4 and S5). Additionally, our inference suggests that rates of recombination are slightly elevated on spike subunit S1 compared to subunit S2 (Fig. S4). If we track recombination events ancestral to the SARS-CoV-2 lineage that are inferred to have happened in the last 100 years, we find evidence for recombination breakpoints occurring close to the 5' end of the spike, just outside the coding region (see Fig. S5). Additionally, we find support for recombination breakpoints toward the 3' end of the spike, near the nucleocapsid gene (see Fig. S5). If we assume that during genome replication in coronaviruses template shifts occur randomly on the genome[10], differences in observed recombination rates could be explained by selection favoring recombinant lineages with breakpoints on 3' to ORF1ab relative to elsewhere on the genome.

We next investigate when different viruses last shared a common ancestor (MRCA) along the genome (see Figs. 1B and S6). RmYN02[33] shares the MRCA with SARS-CoV-2 on the part of the genome that codes for ORF1ab (Fig. 1B). We additionally find strong evidence for one or more recombination events in the ancestry of RmYN02 at the beginning of spike (Fig. 1B). This recent recombination event is unlikely to have occurred with a recent ancestor of any of the coronaviruses included in this dataset since the common ancestor of RmYN02 with any other virus in the dataset is approximately the same (Fig. S6A). In other words, large parts of the spike protein of RmYN02 are as related to SARS-CoV-2 as SARS-CoV-2 is to SARS-CoV-1. The common ancestor timings of P2S across the genome are equal between

RaTG13 and SARS-CoV-2 (Fig. S6C). RaTG13 on the other hand is more closely related to SARS-CoV-2 than P2S (Fig. S6B) across the entire genome.

When looking at when different viruses last shared a common ancestor anywhere on the genome (in other words: when the ancestral lineages of two viruses last crossed paths), we find that RmYN02 has the most recent MRCA with SARS-CoV-2 (Fig. S6C). The median estimate of the most recent MRCA between SARS-CoV-2 and RmYN02 is 1986 (95% CI: 1973–2005). For RaTG13 it is 1975 (95% CI: 1988–1964), for P2S it is 1949 (95% CI: 1907–1973) and with SARS-CoV-1 it is 1834 (95% CI: 1707–1935). These estimates are contingent on a fixed evolutionary rate of $5 \times 10^{-4}$ per nucleotide per year.

**Rates of recombination are associated with rates of adaptation in human seasonal coronaviruses.** We next investigate recombination patterns in MERS-CoV, which has over 2500 confirmed cases in humans, as well as in human seasonal coronaviruses 229E, OC43, and NL63, which have widespread seasonal circulation in humans. As for the SARS-like viruses, we jointly infer recombination networks, rates of recombination, and population sizes for these viruses. We assumed that the genomes evolved under a GTR + $\Gamma_4$ model and, in contrast to the analysis of SARS-like viruses, inferred the evolutionary rates. We observe frequent recombination in the history of all four viruses, wherein genetic ancestry is described by network rather than a strictly branching phylogeny (Fig. 2A–D and Fig. S6A).

The human seasonal coronaviruses all have recombination rates around $1 \times 10^{-5}$ per site and year (Fig. S7). This is around 10–20 times lower than the evolutionary rate (Fig. S8). In contrast to the recombination rates, the evolutionary rates vary greatly across the human seasonal coronaviruses, with rates between a median of $1.3 \times 10^{-4}$ (95% highest posterior density interval (HPD) $1.1-1.5 \times 10^{-4}$) for NL63 and a median rate of $2.5 \times 10^{-4}$ (95% HPD $2.2-2.7 \times 10^{-4}$) and $2.1 \times 10^{-4}$ (95% HPD $1.9-2.3 \times 10^{-4}$) for 229E and OC43 (Fig. S8). These evolutionary

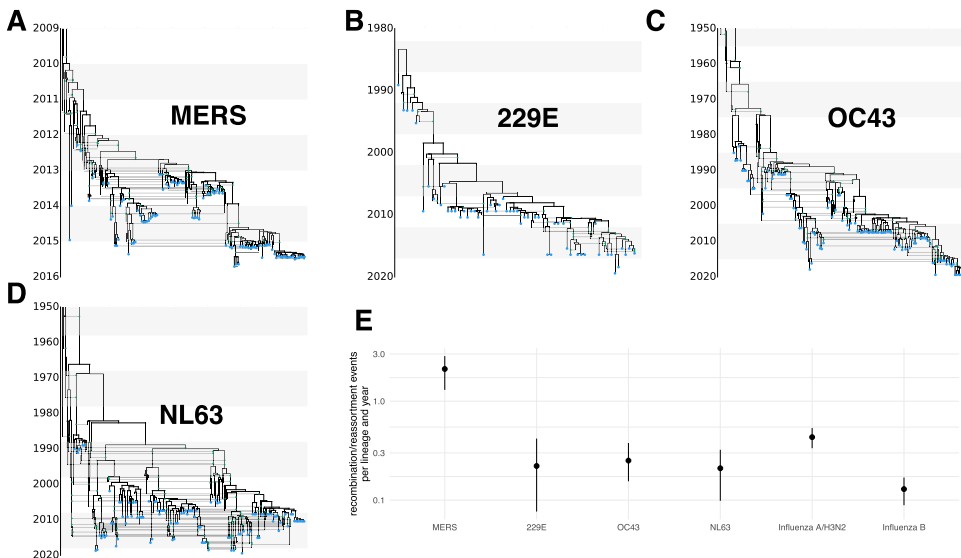

**Fig. 2 Recombination networks and rates for coronaviruses MERS, 229E, OC43, and NL63.** Recombination networks for MERS (**A**) and seasonal human coronaviruses 229E (**B**), OC43 (**C**), and NL63 (**D**). **E** Recombination rates (per lineage and year) for the different coronaviruses compared to reassortment rates in seasonal human influenza A/H3N2 and influenza B viruses as estimated in under the coalescent with reassortment using whole-genome influenza sequences sampled over multiple decades[16]. For OC43 and NL63, the parts of the recombination networks that stretch beyond 1950 are not shown to increase the readability of more recent parts of the networks. The error bars denote the upper and lower bound of the 95% highest posterior density interval. All MCC networks are provided as a Source Data file.

rates are substantially lower than those estimated for SARS-CoV-2 ($1.1 \times 10^{-3}$ substitutions per site and year[35]), which are more in line with our estimates for the evolutionary rates of MERS with a median rate of $6.9 \times 10^{-4}$ (95% HPD $6.0-7.9 \times 10^{-4}$). Evolutionary rate estimates can be time-dependent, with datasets spanning more time estimating lower rates of evolution than those spanning less time[36]. In turn, this means that the evolutionary rate estimates for SARS-CoV-2 will likely be lower the more time passes. It is unclear though if it will approximate the evolutionary rates of other seasonal coronaviruses in the long run.

On a per-lineage basis, the estimated recombination rate for seasonal coronaviruses translates into around 0.1–0.3 recombination events per lineage and year (Fig. 2E). Recombination events defined here are a product of co-infection, recombination, and selection of recombinant viruses. Interestingly, the rate at which recombination events occur is highly similar to the rate at which reassortment events occur in human influenza viruses (Fig. 2D, and ref. [16]). If we assume similar selection pressures for recombinant coronaviruses compared to reassortant influenza viruses, this would indicate similar co-infection rates in influenza and coronaviruses. The incidence of coronaviruses in patients with respiratory illness cases over 12 seasons in western Scotland has been found to be lower (7–17%) than for influenza viruses (13–34%) but to be of the same order of magnitude[37]. Considering that seasonal coronaviruses typically are less symptomatic than influenza viruses, it is not unreasonable to assume that annual incidence, and therefore likely the annual co-infection rates, are comparable between influenza and coronaviruses.

Compared to human seasonal coronaviruses, recombination occurs around 3 times more often for MERS-CoV (Fig. 2E). MERS-CoV mainly circulates in camels and occasionally spills over into humans[38]. MERS-CoV infections are highly prevalent in camels, with close to 100% of adult camels showing antibodies against MERS-CoV[39]. Higher incidence, and thus higher rates of co-infection, could therefore account for higher rates of recombination in MERS-CoV compared to the human seasonal coronaviruses.

We next tested whether parts of the genome with higher rates of recombination are also associated with higher rates of adaptation. To do so, we allowed for different relative rates of recombination within the region 5' of the spike (i.e. mostly ORF1ab), spike itself, and everything 3' of the spike. We computed recombination rate ratios on each of these three sections of the genome as the recombination rate on that section divided by the mean rate on the other two sections. We infer that recombination rates are elevated in the spike protein of all human seasonal coronaviruses considered here (Fig. 3, Figs. S9, and S10). This is consistent with other work estimating higher rates of recombination on the spike protein of betacoronaviruses[11].

We then computed the rates of adaption on different parts of the genomes of the seasonal human coronaviruses using the approach described in refs. [40,41]. This approach does not explicitly consider trees to compute the rates of adaptation on different parts of the genomes and is not affected by recombination[41]. We find that sections of the genome with relatively higher rates of adaptation correspond to sections of the genome with relatively higher rates of recombination (Fig. 3). In particular, recombination and adaptation are elevated on the section of the genome that codes for the spike protein and are lower elsewhere.

We next investigated whether these trends hold when looking only at spikes. The spike protein is made up of two subunits: S1 and S2. S1 binds to the host cell receptor, while S2 facilitates fusion of the viral and cellular membrane[42]. Rates of adaptation have been shown to be high in S1, but not S2, for 229E and OC43[41]. While the rates of adaptation are relatively low overall for NL63, there is still some evidence that they are elevated in S1 compared to S2[41].

To test whether recombination rates vary with rates of adaptation on the subunits of the spike as well, we inferred the recombination rates from the spike only, allowing for different rates of recombination on S1 versus the rest of the spike. We find that the rates of recombination are elevated on S1 for 229E and OC43 compared to the rest of the spike gene (Fig. 3). This is consistent with strong absolute rates of adaptation on S1 on these

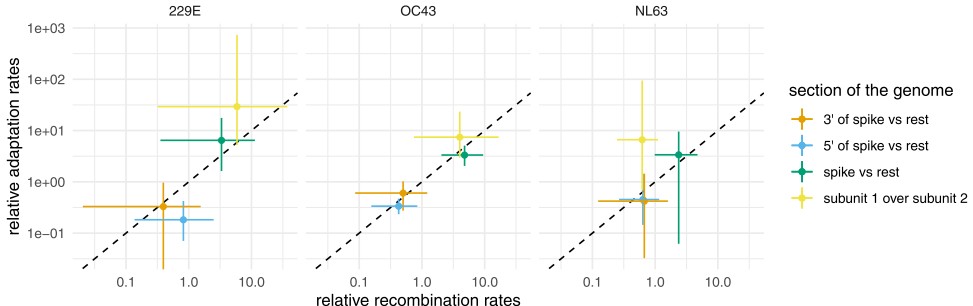

**Fig. 3 Comparison of recombination rates with rates of adaptation on different parts of the genomes of seasonal human coronaviruses 229E, OC43, and NL63.** Association between estimated relative recombination rate (*x*-axis) and relative adaptation rate (*y*-axis) for three different seasonal human coronaviruses: 229E, OC43, and NL63. These estimates are shown for different parts of the genome, indicated by the different colors. These results from two different types of analysis: one using spike only (subunit 1 over subunit 2, shown in yellow) and one using the full genome (shown in orange, blue, and green). The rate ratios denote the rate on a part of the genome divided by the average rate on the two other parts of the genome. The error bars of the recombination rates (*x*-axis) denote the upper and lower bounds of the 95% HPD intervals of the estimates of relative recombination rates. The error bars of the rates of adaptation are computed using 100 bootstrapped outgroups and alignments when computing the rates of adaptation. Source data are provided as a Source Data file.

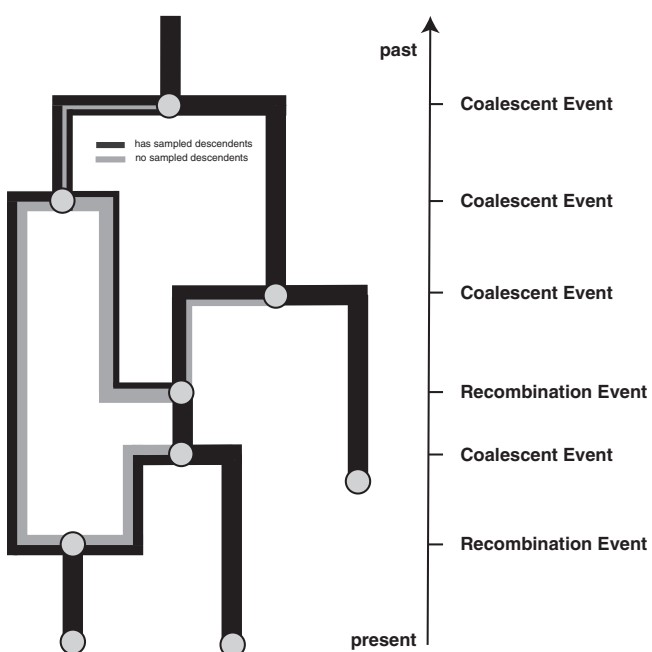

**Fig. 4 Example recombination network.** Events that can occur on a recombination network as considered here. We consider events to occur from the present backward in time to the past (as is the norm when looking at coalescent processes). Lineages can be added upon sampling events, which occur at predefined points in time and are conditioned on. Recombination events split the path of a lineage in two, with everything on one side of a recombination breakpoint going in one direction and everything on the other side of a breakpoint going in the other direction.

two viruses. For NL63, we find weak evidence for the rate on S2 to be slightly higher than on S1 (Fig. 3), even though the rates of adaptation are inferred to be higher on S1. The absolute rate of adaptation in S1 of NL63 is, however, substantially lower than for 229E or OC43. Additionally, the uncertainty around the estimates on adaption rate ratios between the two subunits for NL63 is rather large and includes no difference at all. Overall, these results suggest that particular recombination events that have resulted in recombinant viruses are either positively or negatively selected. Elevated rates of recombination in areas where adaptation is stronger have been described for other organisms (reviewed here[43]). Alternatively, higher rates of recombination could also be

due to mechanistic reasons, as has been suggested in the case of SARS-CoV-2[44].

To further investigate this, we next computed the rates of recombination on fitter and less fit parts of the recombination networks of 229E, OC43, and NL63. To do so, we first classify each edge of the inferred posterior distribution of the recombination networks into fit and unfit based on how long a lineage survives into the future. Fit edges are those that have descendants at least 1, 2, 5, or 10 years into the future, and unfit edges are those that do not. We then computed the rates of recombination on both types of edges for the entire posterior distribution of networks. Overall, we do not find that fit edges show relatively higher rates of recombination (see Fig. S11). The simplest explanation is that we do not have enough data points to measure recombination rates on unfit edges, meaning to measure recombination rates on part of the recombination network where selection had too little time to shape which lineages survive and which go extinct. An alternative explanation to why we see elevated rate or recombination in the spike protein, but do not observe a population level fitness benefit could be that most (outside of spike) recombinants could be detrimental to fitness with few (within spike) having little fitness effect at all.

## Discussion

Though not yet highly prevalent, evidence for recombination in SARS-CoV-2 has started to appear [45–48]. As such, it is crucial to know the extent to which recombination is expected to shape SARS-CoV-2 in the coming years, to have methods to identify recombination, and to perform phylogenetic reconstruction in the presence of recombination. The results shown here indicate that some recombinant viruses are either positively or negatively selected. Estimating the deleterious load of viruses before and after recombination using ancestral sequence reconstruction[49] could help shed light on which sequences are favored during recombination. Furthermore, having additional sequences to reconstruct recombination patterns in the seasonal coronaviruses should clarify the role recombination plays in the long-term evolution of these viruses.

While their impact on the evolutionary dynamics of SARS-CoV-2 remains unclear, the likely rise of future SARS-CoV-2 recombinants will further necessitate methods that allow phylogenetic and phylodynamic inferences to be performed in the presence of recombination[50]. In absence of that, recombination has to be either ignored, leading to biased phylogenetic and phylodynamic reconstruction[15], or non-recombinant parts of the

genome have to be used for analyses, reducing the precision of these methods. Our approach addresses this gap by providing a Bayesian framework to infer recombination networks. To facilitate easy adaptation, we implemented the method so that analyses can be set up following the same workflow as regular BEAST2[27] analyses. Extending the current suite of population dynamic models, such as birth–death models[51] or models that account for population structure[52,53], will further increase the applicability of recombination models to study the spread of pathogens.

## Methods

**Coalescent with recombination**. The coalescent with recombination models a backward in time coalescent and recombination process[17]. In this process, three different events are possible: sampling, coalescence, and recombination. Sampling events happen at predefined points in time. Recombination events happen at a rate proportional to the number of coexisting lineages at any point in time. Recombination events split the path of a lineage in two, with everything on one side of a recombination breakpoint going in one ancestral direction and everything on the other side of a breakpoint going in the other direction. As shown in Fig. 4, the two parental lineages after a recombination event each carry a subset of the genome. In reality, the viruses corresponding to those two lineages still carry the full genome, but only a part of it will have sampled descendants. In other words, only a part of the genome carried by a lineage at any time may impact the genome of a future lineage that is sampled. The probability of actually observing a recombination event on lineage $l$ is proportional to how much genetic material that lineage carries. This can be computed as the difference between the last and first nucleotide position that is carried by $l$, which we denote as $\mathcal{L}(l)$. Coalescent events happen between co-existing lineages at a rate proportional to the number of pairs of coexisting lineages at any point in time and inversely proportional to the effective population size. The parent lineage at each coalescent event will carry genetic material corresponding to the union of the genetic material of the two-child lineages.

**Posterior probability**. In order to perform joint Bayesian inference of recombination networks together with the parameters of the associated models, we use a MCMC algorithm to characterize the joint posterior density. The posterior density is denoted as:

$$P(N, \mu, \theta, \rho | D) = \frac{P(D|N, \mu)P(N|\theta, \rho)P(\mu, \theta, \rho)}{P(D)}, \quad (1)$$

where $N$ denotes the recombination network, $\mu$ the evolutionary model, $\theta$ the effective population size and $\rho$ the recombination rate. The multiple sequence alignment, that is the data, is denoted $D$. $P(D|N, \mu)$ denotes the network likelihood, $P(N|\theta, \rho)$, the network prior and $P(\mu, \theta, \rho)$ the parameter priors. As is usually done in Bayesian phylogenetics, we assume that $P(\mu, \theta, \rho) = P(\mu)P(\theta)P(\rho)$.

Using a Bayesian approach has several advantages. In particular, it allows us to account for uncertainty in the parameter and network estimates. Additionally, it allows balancing different sources of information against each other. The coalescent with recombination model, for example, will tend to favor networks with fewer recombination events. The cost of adding more recombination events depends on the recombination rate. At lower rates of recombination, adding new recombination events is more costly and the information coming from the sequence alignment in support of a recombination event needs to be greater.

**Network likelihood**. While the evolutionary history of the entire genome is a network, the evolutionary history of each individual position in the genome can be described as a tree. We can therefore denote the likelihood of observing a sequence alignment (the data denoted $D$) given a network $N$ and evolutionary model $\mu$ as

$$P(D|N, \mu) = \prod_{i=1}^{\text{sequence length}} P(D_i|T_i, \mu), \quad (2)$$

with $D_i$ denoting the nucleotides at position $i$ in the sequence alignment and $T_i$ denoting the tree at position $i$. The likelihood at each individual position in the alignment can then be computed using the standard pruning algorithm[54]. We implemented the network likelihood calculation $P(D_i|T_i, \mu)$ such that it allows making use of all the standard site models in BEAST2. Currently, we only consider strict clock models and therefore do not allow for rate variations across different branches of the network. This is because the number of edges in the network changes over the course of the MCMC, making relaxed clock models more complex to implement. We implemented the network likelihood such that it can make use of caching of intermediate results and use unique patterns in the multiple sequence alignment, similar to what is done for tree likelihood computations.

**Network prior**. The network prior is denoted by $P(N|\theta, \rho)$, which is the probability of observing a network and the embedding of segment trees under the coalescent

with recombination model, with effective population size $\theta$ and per-lineage recombination rate $\rho$. It plays essentially the same role that tree prior plays in phylodynamic analyses on trees.

We can calculate $P(N|\theta, \rho)$ by expressing it as the product of exponential waiting times between events (i.e., recombination, coalescent, and sampling events):

$$P(N|\theta, \rho) = \prod_{i=1}^{\#events} P(event_i|L_i, \theta, \rho) \times P(interval_i|L_i, \theta, \rho), \quad (3)$$

where we define $t_i$ to be the time of the $i$th event and $L_i$ to be the set of lineages extant immediately prior to this event. (That is, $L_i = L_t$ for $t \in [t_{i-1}, t_i)$.

Given that the coalescent process is a constant size coalescent and given the $i$th event is a coalescent event, the event contribution is denoted as

$$P(event_i|L_i, \theta, \rho) = \frac{1}{\theta}. \quad (4)$$

If the $i$th event is a recombination event and assuming constant rates of recombination over time, the event contribution is denoted as

$$P(event_i|L_i, \theta, \rho) = \rho * \mathcal{L}(l). \quad (5)$$

The interval contribution denotes the probability of not observing any event in a given interval. It can be computed as the product of not observing any coalescent, nor any recombination events in interval $i$. We can therefore write:

$$P(interval_i|L_i, \theta, \rho) = \exp[-(\lambda^c + \lambda^r)(t_i - t_{i-1})], \quad (6)$$

where $\lambda^c$ denotes the rate of coalescence and can be expressed as

$$\lambda^c = \binom{|L_i|}{2} \frac{1}{\theta}, \quad (7)$$

and $\lambda^r$ denotes the rate of observing a recombination event on any co-existing lineage and can be expressed as

$$\lambda^r = \rho \sum_{l \in L_i} \mathcal{L}(l). \quad (8)$$

In order to allow the recombination rates to vary across $s$ sections $\mathcal{S}_s$ on the genome, we modify $\lambda^r$ to differ in each section $\mathcal{S}_s$, such that:

$$\lambda^r = \sum_{s \in \mathcal{S}} \rho_s \sum_{l \in L_i} \mathcal{L}(l) \cap \mathcal{S}_s, \quad (9)$$

with $\mathcal{L}(l) \cap \mathcal{S}_s$ denoting the amount of overlap between $\mathcal{L}(l)$ and $\mathcal{S}_s$. The recombination rate in each section $s$ is denoted as $\rho_s$.

**MCMC algorithm for recombination networks**. In order to explore the posterior space of recombination networks, we implemented a series of MCMC operators. These operators often have analogs in operators used to explore different phylogenetic trees and are similar to the ones used to explore reassortment networks[16]. Here, we briefly summarize each of these operators.

*Add/remove operator*: The add/remove operator adds and removes recombination events. An extension of the subtree prune and regraft move for networks[55] to jointly operate on segment trees as well. We additionally implemented an adapted version to sample re-attachment under a coalescent distribution to increase acceptance probabilities.

*Loci diversion operator*: The loci diversion operator randomly changes the location of recombination breakpoints of a recombination event.

*Exchange operator*: The exchange operator changes the attachment of edges in the network while keeping the network length constant.

*Subnetwork slide operator*: The subnetwork slide operator changes the height of nodes in the network while allowing to change in the topology.

*Scale operator*: The scale operator scales the heights of the root node or the whole network without changing the network topology.

*Gibbs operator*: The Gibbs operator efficiently samples any part of the network that is older than the root of any segment of the alignment and is thus not informed by any genetic data and is the analog to the Gibbs operator in[16] for reassortment networks.

*Empty loci preoperator:* The empty loci preoperator augments the network with edges that do not carry any loci for the duration of one of the above moves, to allow for larger jumps in network space.

One of the issues when inferring these recombination networks is that the root height can be substantially larger than when not allowing for recombination events. This can cause a computational issue when performing inferences. To circumvent this, we truncate the recombination networks by reducing the recombination rate sometime after all positions of the sequence alignment have reached their common ancestor height.

*Validation and testing*. We validate the implementation of the coalescent with recombination network prior as well as all operators in Fig. S12. We also show that truncating the recombination networks does not affect the sampling of recombination networks prior to reaching the common ancestor height of all positions in the sequence alignment.

We then tested whether we are able to infer recombination networks, recombination rates, effective population sizes, and evolutionary parameters from

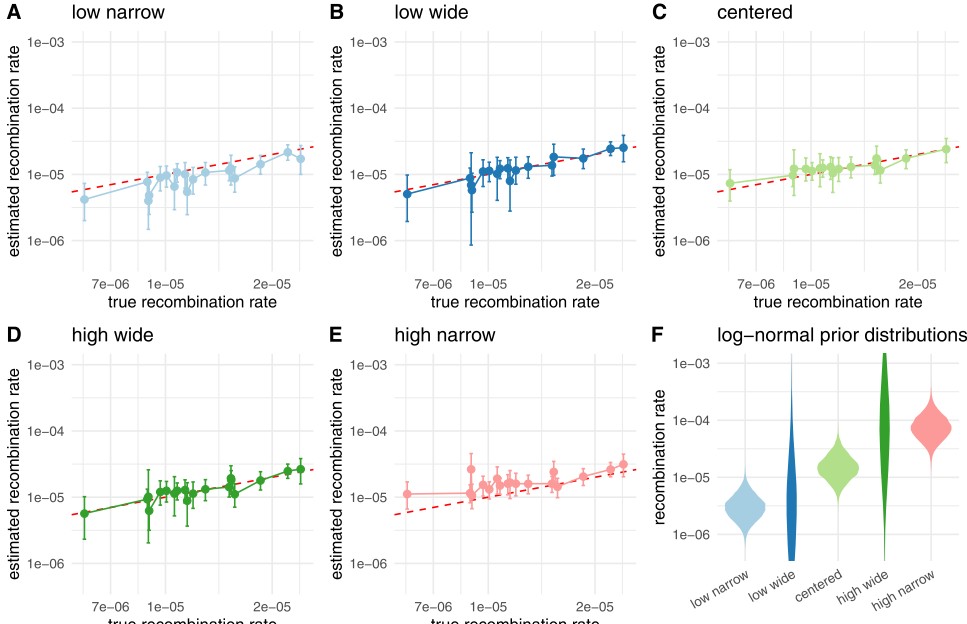

**Fig. 5 Impact of the recombination rate prior distribution on the inferred recombination rates.** Here, we compare then inferred recombination rates when using different prior distributions that differed from the distributions from which the rates for simulations were sampled. The rates for simulations were sampled from a log-normal distribution with $\mu = -11.12$ and $\sigma = 0.5$. In **A**, we show the inferred rates when using a prior distribution with $\mu = -12.74$ and $\sigma = 0.5$ (leading to a 5 times lower mean in real space than the correct prior). In **B**, we show the inferred rates when using a prior distribution with $\mu = -12.74$ and $\sigma = 2$. In **C**, we show the inferred rates when using the same prior distribution as was sampled under. In **D**, we show the inferred rates when using a prior distribution with $\mu = -9.72$ and $\sigma = 2$. In **E**, we show the inferred rates when using a prior distribution with $\mu = -9.72$ and $\sigma = 0.5$ (leading to 5 times higher mean in real space than the correct prior). **F** shows the corresponding density plots for all log-normal distributions used as prior distributions on the recombination rates.

simulated data. To do so, we randomly simulated recombination networks under the coalescent with recombination. On top of these, we then simulated multiple sequence alignments. We then re-infer the parameters used to simulate using our MCMC approach. As shown in Fig. S13, these parameters are retrieved well from simulated data with little bias and accurate coverage of simulated parameters by credible intervals.

We next tested how well we can retrieve individual recombination events. To do so, we plot the location and timings of simulated recombination events for the first 9 out of 100 simulations. We then plot the density of recombination events in the posterior distribution of networks, based on the timing and location of the inferred breakpoint on the genome. As shown in Fig. S14, we are able to retrieve the true (simulated) recombination events well.

We next tested how the speed of inference scales with the number of recombination events, the number of samples in the dataset, and the evolutionary rate. To do so, we simulated 300 recombination networks and sequence alignment of length 10,000 under a Jukes–Cantor model with between 10 and 200 leaves and a recombination rate between $1 \times 10^{-5}$ and $2 \times 10^{-5}$ recombination events per site per year. This means that for each simulation, there were between 0 and 100 recombination events, allowing us to investigate how the inference scales in different settings. As shown in Fig. S15, the ESS per hour decreases with the number of recombination events and samples, but not the evolutionary rates. In particular, the ESS per hour decreases much faster with the number of recombination events in a dataset than the number of samples. This suggests that the methods can currently be used more easily to analyze a dataset with a large number of samples over a large number of recombination events.

We next tested how the choice of the prior distribution on the recombination rate impacts the recombination rate estimate. To do so, we simulate 20 recombination networks and sequence alignment of length 10,000 under a Jukes–Cantor model with 100 leaves and a recombination rate drawn randomly from a log-normal distribution. We then infer the recombination rates using 5 different recombination rate priors as shown in Fig. 5F that put some or a lot of weight on the wrong parameters. As shown in Fig. 5A–E, we are able to infer recombination rates, even with the wrong priors.

Additionally, we compared the effective sample size values from MCMC runs inferring recombination networks for the MERS spike protein to treating the evolutionary histories as trees. We find that although the effective sample size values are lower when inferring recombination networks, they are not orders of magnitude lower (see Fig. S16).

**Recombination network summary**. We implemented an algorithm to summarize distributions of recombination networks similar to the maximum clade credibility framework typically used to summarize trees in BEAST[56]. In short, the algorithm summarizes individual trees at each position in the alignment. To do so, we first compute how often we encountered the same coalescent event at every position in the alignment during the MCMC. We then choose the network that maximizes the clade support over each position as the maximum clade credibility (MCC) network.

The MCC networks are logged in the extended Newick format[57] and can be visualized in icytree.org[58]. We here plotted the MCC networks using an adapted version of baltic (https://github.com/evogytis/baltic).

**Sequence data**. The genetic sequence data for OC43, NL63, and 229E were obtained from ViPR (http://www.viprbrc.org) and were the same as used[41]. All these sequences were isolated from a human host and downsampled from the dataset used in ref. [41] to 100 sequences (for OC43 and NL63). As there were only 54 229E sequences, we did not do any downsampling on this data. The sequence data for the MERS analyses were the same as described in ref. [38], but using a randomly down sampled dataset of 100 sequences. For the SARS-like analyses, we used 40 different deposited SARS-like genomes, mostly originating from bats, as well as humans, and one pangolin-derived sequence.

**Rates of adaptation**. The rates of adaptation were calculated using a modification of the McDonald–Kreitman method, as designed by Bhatt et al.[40], and implemented in ref. [41]. Briefly, for each virus, we aligned the sequence of each gene or genomic region. Then, we split the alignment into 3-year sliding windows, each containing a minimum of 3 sequenced isolates. We used the consensus sequence at the first time point as the outgroup. A comparison of the outgroup to the alignment of each subsequent temporal yielded a measure of synonymous and non-synonymous fixations and polymorphisms at each position in the alignment. This approach requires having sequence data gathered over relatively long time periods where the consensus genome allows for an accurate description of the long-term evolutionary patterns and, as such, would not be adequate for a pathogen with a relatively short evolutionary history, such as for SARS-CoV-2. We used proportional site counting for these estimations[59]. We assumed that selectively neutral sites are all silent mutations as well as replacement polymorphisms occurring at frequencies between 0.15 and 0.75[40]. We identified adaptive substitutions as non-synonymous fixations and high-frequency polymorphisms that exceed the neutral expectation. We then estimated the rate of adaptation (per codon per year) using linear regression of the number of adaptive substitutions inferred at each time point. In order to compute the 5' spike and 3' spike rates of adaptation, we used the

weighted average of all coding regions to the left (upstream) or right (downstream) of the spike gene, respectively, using the length of the individual sections as weights. We estimated the uncertainty by running the same analysis on 100 bootstrapped outgroups and alignments.

**Reporting summary**. Further information on research design is available in the Nature Research Reporting Summary linked to this article.

## Data availability

The BEAST2 input xml files for all coronavirus analyses in this manuscript, as well as the files used to post process these analyses are available from https://github.com/nicfel/Recombination-Material and here ref. [60]. The xml files include the sequence data and exact input specification of the coronavirus analyses performed in this manuscript, except for the sequences published on gisaid. The acknowledgment table for the four gisaid sequences used for the SARS-like analyses is provided in Supplementary Note 1. The genbank accession numbers for the 229E, OC43, NL63, SARS-like, and MERS analyses are provided as separate tables in Supplementary Data 1. The MERS sequences without accession numbers are used from ref. [38]. Source data are provided with this paper.

## Code availability

The Recombination package is implemented as an addon to the Bayesian phylogenetics software platform BEAST2[27]. All MCMC analyses performed here were run using adaptive parallel tempering[61]. The source code is available at https://github.com/nicfel/Recombination and here ref. [62]. We additionally provide a tutorial on how to set up and post-process analysis at https://github.com/nicfel/Recombination-Tutorial. The MCC networks are plotted using an adapted version of baltic (https://github.com/evogytis/baltic). All other plots are done in R using ggplot2[63] and ggenes[64].

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

## Acknowledgements
We would like to thank Timothy G. Vaughan for his helpful insights into the implementation of the software. N.F.M. is funded by the Swiss National Science Foundation (P2EZP3_191891). K.E.K. is a NSF GRFP Fellow (DGE-1762114). T.B. is a Pew Biomedical Scholar and is supported by NIH R35 GM119774. The Scientific Computing Infrastructure at Fred Hutch is supported by NIH ORIP S10OD028685.

## Author contributions
N.F.M. and T.B. conceived and designed the experiments. N.F.M. and K.E.K. performed the statistical analysis and analyzed the data. N.F.M. implemented the software. N.F.M., K.E.K., and T.B. wrote the paper.

## Competing interests
The authors declare no competing interests.
