## [Peer Review File · Nature Communications]

A Bayesian approach to infer recombination patterns in coronavirusesReviewers' Comments:

Reviewer #1:

Remarks to the Author:

I was really looking forward to reading this manuscript, but I don't think it's clear enough in its presentation to be published at the moment. I do believe that recombination in coronaviruses is common, but I don't think the approach in this ms is the best way to describe recombination history in the four chosen coronaviruses.

1. In the coalescent-with-recombination, is there any cost to recombination? Can lineages continue to split in the past indefinitely? How costly is a homoplasy compared to a recombination event in this implementation? A mutation occurs with a probability described in the GTR+Gamma model, and the parameters in this model are fit to the data (assuming no recombination, I suppose) but how is the recombination parameter fit? For example, if the GTR-model allows you to account for ten A->C mutations with an explicit probability that has been fit, but a recombination event allows you to replace all ten of these mutations with a single recombination event, where is the comparison relating the (1) recombination rate to the (2) GTR-model probabilities for all ten A->C mutations? What if it's three A->C mutations? What if it's 100 A->C mutations? Surely, the probabilities should favor recombination if it's 100 A->C mutations but not if it's three.

2. Figures 1 and 2 look beautiful. But you can't see what any of the recombination events are. And you can't see how many there are.

3. Figure 4 is very clear. Can some of the recombination events be drawn like this, with larger font and timing of when specific lineage may have been formed by recombination?

4. The authors say that there is no recombination with distantly related viruses. Then, in their analysis, what is the history of the Zhejiang viruses (ZC45 and ZXC21) that have appeared as recombinants in other published papers?

5. What is the 2×10^{-6} recombination rate? Per site per year? Per site per replication? Per genome per replication? Per pathogen-generation time per replication? If it's per site per year (as seems to be the case based on other parts of the paper) that means a SARS-like lineage should have a recombination event in the past on average every 15 years. Is that consistent with Figure 1?

6. Figure S9A is great, and is the starting point I would use to begin re-working parts of this paper. Can the authors take a data set of the same size and diversity as Figure 1, simulate it with a handful of very rare recombination rates, and see if they can reconstruct a few of the most recent recombination events correctly? No need to make the y-axis 2000 years in this analysis. Just make the y-axis 100 years, and see if the first 2 or 3 recombination events (i.e., the most recent ones) are reconstructed correctly.

7. Abstract: recombination can be beneficial or detrimental to fitness. It can put beneficial mutations together or break them apart.

Reviewer #2:

Remarks to the Author:

This is a very interesting paper that centers on the important issue of recombination in SARS-CoVs and the development of appropriate phylogeny inference methods that can accurately investigate and represent sequence datasets where the assumption of shared ancestry is violated. The topic is central to our understanding of coronavirus evolution, and Bayesian coalescence methods that take explicitly into account recombination are indeed urgently needed. Unfortunately, the manuscript also suffers of

a few methodological problems, and relies on unconvincing assumptions that must be addressed by the authors to strengthen confidence in the reliability of their work and results.

Major remarks:

- SARS-like viruses recombination rate estimated by the authors with their new method, 2×10^{-6} , is significantly lower than rates observed in other recombining RNA viruses. For HIV, for example, recombination rate, in case of co-infection, is actually higher than mutation rate. This begs the question of whether the estimated rate is the result of a bias in their method leading to a systematic error. The authors need to provide a set of simulations using a range of recombination rates that would allow to validating their method and exploring its potential limitations.

- The authors also report that recombination rate seems elevated in the Spike protein and suggest that this could be explained by selection favoring recombination events 3' to ORF1ab. This may be true, but the overall discussion on how selection has impacted recombination is quite superficial and seems to ignore some of the recent literature. Tagliamonte et al., for example, have shown how interplay between recombination and negative selection in SARS-CoV evolution have both played a role in increasing covariant dynamic movements of the Spike glycoprotein, while preserving inter-domain structure.

- Rates of adaptation are estimated using mainly the method implemented by Kistler and Bedford (2021) by modifying Bhatt's work. This is a site-counting method that estimates the product between a fixation or polymorphism score and a silent or replacement score, and eventually sums up the products for all sites belonging to a specific frequency class. The method is problematic in the context of the authors' study since fixation and polymorphism scores depend on the number of different nucleotides observed at a site, which in turn likely depends on the sampling size of the population under study. Given the disproportionate amount of SARS-CoV-2 sequence data available, for example, any analysis must select smaller datasets that may or may not be fully representative of the underlying populations, with the unavoidable result of over or under estimating the number of transient polymorphisms, especially when using a method that is not based on phylogeny and cannot distinguish between substitutions occurring on internal (more likely to be fixed) or terminal (more likely to be transient) branches of the tree.

- A paper that proposes a new Bayesian phylogeny method must have a section evaluating performance when analyzing increasingly large sequence data set, computational cost and feasibility, as well exploring potential problems introduced by the often arbitrary choice of priors. None of this is discussed in the present manuscript, which is in my opinion a major flaw. However, I am confident the authors could easily address these points.

Minor remarks:

- Line 40 "Recombination poses a unique challenge phylogenetic methods" should be corrected into "Recombination poses a unique challenge to phylogenetic methods".

- Line 86 "A fixed rate of rate of 5×10^{-4} per nucleotide and year" should be replaced by "per nucleotide per year".

- Line 88 "This means that the sampling times themselves therefore offers, therefore little information" the word "therefore" should be removed.

- The fixed evolutionary rate chosen by the authors seems reasonable, but I wonder if it would be more appropriate to use a lognormal distribution with confidence intervals matching the uncertainty of current estimates.

- Many Network-based methods, not necessarily based on Bayesian phylogenetics, have been developed and proposed in the literature to investigate and display shared ancestry, such as split decomposition methods. The most relevant should be at least quoted in the introduction to provide a better context to the readers.

Reviewer #3:

Remarks to the Author:

Muller et al present an MCMC method for detecting recombination in virus datasets and apply it to coronaviruses. This approach infers the underlying reticulate history in the alignment, i.e. explicitly allows for the alternative phylogenetic histories that are taking place. The method is sound and builds an existing theoretical framework. The authors apply their method to several coronavirus data sets, in particular SARS-CoV-2. These results are interesting. Some comments:

Line 23, the Andersen proximal paper addressed whether SCV2 was likely to be natural versus lab-made so not really its origin specifically.

Line 25 -- Voltz's phylodynamic methods assess virus growth rates not fitness explicitly.

Line 29 -- mutations are also introduced by anti-viral molecules eg APOBEC, not just replication errors.

Line 78 -- Boni et al. 2020, <https://www.nature.com/articles/s41564-020-0771-4>. show recombination was probably not the origin of SARS-CoV-2 RBM.

We would expect recombination rates to be higher proximal to the regions of the virus genome under selection. Is this what the authors mean when they say these sections have elevated rates? This conflicts with a reporting of a lack of recombination breakpoints in S1 - <https://www.biorxiv.org/content/10.1101/2021.01.22.427830v3>.

On line 156 the authors write that the evolutionary purpose of recombination is not known but later discuss classical explanations by Barton, Hill & Robertson etc. Please be consistent on this important literature and discuss in the introduction.

The authors should check the SARS-CoV-2 literature for other papers on recombination eg:

<https://www.biorxiv.org/content/10.1101/2020.09.21.300913v2>

<https://www.medrxiv.org/content/10.1101/2021.06.18.21258689v1>

<https://www.biorxiv.org/content/10.1101/2021.01.21.427579v3.full>

<https://www.biorxiv.org/content/10.1101/2020.03.16.993816v2>

<https://www.biorxiv.org/content/10.1101/2020.02.10.942748v3>

Given the interest in SARS-CoV-2's ongoing evolution and escape from vaccines, do the authors predict a role for recombination?

We would like to thank all the reviewers for taking time to review this manuscript and for their helpful comments that improved the manuscript. We addressed all the points raised individually below. In particular, we added a novel way to represent recombination events in space (location on the genome) and time, as well as improved the simulation study, to investigate the performance of the method more in depth.

REVIEWER COMMENTS

Reviewer #1 (Remarks to the Author):

I was really looking forward to reading this manuscript, but I don't think it's clear enough in its presentation to be published at the moment. I do believe that recombination in coronaviruses is common, but I don't think the approach in this ms is the best way to describe recombination history in the four chosen coronaviruses.

1. In the coalescent-with-recombination, is there any cost to recombination? Can lineages continue to split in the past indefinitely? How costly is a homoplasy compared to a recombination event in this implementation? A mutation occurs with a probability described in the GTR+Gamma model, and the parameters in this model are fit to the data (assuming no recombination, I suppose) but how is the recombination parameter fit? For example, if the GTR-model allows you to account for ten A->C mutations with an explicit probability that has been fit, but a recombination event allows you to replace all ten of these mutations with a single recombination event, where is the comparison relating the (1) recombination rate to the (2) GTR-model probabilities for all ten A->C mutations? What if it's three A->C mutations? What if it's 100 A->C mutations? Surely, the probabilities should favor recombination if it's 100 A->C mutations but not if it's three.

The cost to recombination comes from the coalescent with recombination model that is essentially a network prior (analogue to the tree prior for phylogenetic tree analyses). In this model each recombination event has a "cost" in that an event occurs which will be penalized via likelihood based on the recombination rate parameter. Just as likelihood will generally favor a more parsimonious number of substitutions along a phylogeny branch (while permitting more substitutions with lower likelihood), the coalescent with recombination model will favor a recombination network that is more parsimonious.

Depending on that network prior, more or less information from the genetic data is needed for recombination events to be supported by the inference. At high rates of recombination, less information is needed for recombination events to be inferred. This is because the coalescent with recombination model in this case a priori assumes that recombination events are more likely. At lower rates, the opposite is true, where the model a priori assume that recombination events are less likely to occur and will therefore need more "convincing" by the data to support a recombination events To address this, we now added more detail in the introduction on how the method balances different information streams. In particular, we write in the introduction that: "Using a Bayesian approach has several advantages, in particular, it allows us to account for

uncertainty in the parameter and network estimates. Additionally, it allows balancing different sources of information against each other. The coalescent with recombination model, for example, will tend to favor networks with fewer recombination events. The implicit cost of adding more recombination events itself depends on the recombination rate. At lower rates of recombination, adding new recombination events is more costly and the information coming from the sequence alignment in support of a recombination event needs to be greater.”

2. Figures 1 and 2 look beautiful. But you can't see what any of the recombination events are. And you can't see how many there are.

We now added five new figures (S4, S5, S10, S11 & S14) that denote where (on the genomes) and when (in time) recombination events are inferred to have occurred, both for simulations and the coronavirus analyses. We also did this for recombination events ancestral to SARS-CoV-2 and find support for several events in the last century (S4).

3. Figure 4 is very clear. Can some of the recombination events be drawn like this, with larger font and timing of when specific lineage may have been formed by recombination?

The network plotting we use at the moment rely on essentially plotting trees with each recombination event being displayed as a horizontal edge that attaches to a new leaf. We, unfortunately, don't really have a good way (at the moment) to plot ancestral lineages as seen in Figure 4 other than by hand. As written above, we now added new figures to display when and where along the genome recombination events are estimated to have occurred. Generally, plotting recombination networks when there are many recombination events represents a tricky visualization challenge that we're still trying to figure out how best to deal with. In the manuscript we try to emphasize summary statistics and associated credible intervals where possible.

4. The authors say that there is no recombination with distantly related viruses. Then, in their analysis, what is the history of the Zhejiang viruses (ZC45 and ZXC21) that have appeared as recombinants in other published papers?

The sentence on recombination with distantly related viruses was unclear and we removed it. We analysed (only in the response letter) the history of the two Zhejiang viruses and they do not seem to have a common ancestor time anywhere on the genome with SARS-CoV-2 in the last approx. 100 years.

5. What is the 2×10^{-6} recombination rate? Per site per year? Per site per replication? Per genome per replication? Per pathogen-generation time per replication? If it's per site per year (as seems to be the case based on other parts of the paper) that means a SARS-like lineage should have a recombination event in the past on average every 15 years. Is that consistent with Figure 1?

We now clarify that this is the recombination rate per site and and year and include in the text:

“This recombination rate is a function of co-infection rates, probability of recombination occurring upon co-infection and selection. As such, the recombination rate we infer here will be, possibly substantially, lower than the within host rate of recombination.”

As we show in the newly added Figure S4, we find there to be several recombination events in the last 100 years on the lineage ancestral to SARS-CoV-2, which is broadly consistent with the expectation for the number of events.

6. Figure S9A is great, and is the starting point I would use to begin re-working parts of this paper. Can the authors take a data set of the same size and diversity as Figure 1, simulate it with a handful of very rare recombination rates, and see if they can reconstruct a few of the most recent recombination events correctly? No need to make the y-axis 2000 years in this analysis. Just make the y-axis 100 years, and see if the first 2 or 3 recombination events (i.e., the most recent ones) are reconstructed correctly.

As part of the simulation study, we now show how true recombination events and the distribution of inferred recombination events compare. While we are currently not able to get posterior support values for individual recombination events (which is an area of ongoing research related to the summary of distributions of networks), we find that the distribution of inferred events matches the true events well. This analysis is shown in new Figure S14 and referenced in the manuscript under the Methods section “Validation and testing”. We also added the same type of figure for SARS-CoV-2 (new Figure S4), MERS, OC43, 229E and NL63 (new Figure S10) to display recombination events.

7. Abstract: recombination can be beneficial or detrimental to fitness. It can put beneficial mutations together or break them apart.

In the abstract, “beneficial” specifically refers to the correlation between higher rates of adaptation and higher rates of recombination.

Reviewer #2 (Remarks to the Author):

This is a very interesting paper that centers on the important issue of recombination in SARS-CoVs and the development of appropriate phylogeny inference methods that can accurately investigate and represent sequence datasets where the assumption of shared ancestry is violated. The topic is central to our understanding of coronavirus evolution, and Bayesian coalescence methods that take explicitly into account recombination are indeed urgently needed. Unfortunately, the manuscript also suffers of a few methodological problems, and relies on unconvincing assumptions that must be addressed by the authors to strengthen confidence in the reliability of their work and results.

Major remarks:

- SARS-like viruses recombination rate estimated by the authors with their new method, 2×10^{-6} , is significantly lower than rates observed in other recombining RNA viruses. For HIV, for example, recombination rate, in case of co-infection, is actually higher than mutation rate. This begs the question of whether the estimated rate is the result of a bias in their method leading to a systematic error. The authors need to provide a set of simulations using a range of recombination rates that would allow to validating their method and exploring its potential limitations.

The recombination events we consider here are between different viral lineages and therefore conditional on there being a co-infection event. As such, the recombination rate is not directly the within host recombination rate, but a function of the chance of recombination upon co-infection, the fitness of the recombinants and the rate of co-infection. To put the recombination rate of 2×10^{-6} into context, we now write in the text that:

“This rate translates to about 0.06 recombination events per lineage per year, which is slightly lower than the estimated rate of recombination for the seasonal human coronaviruses and the reassortment rates for pandemic 1918 like influenza A/H1N1 and influenza B viruses, which are all around 0.1-0.2 reassortment events per lineage per year (Mueller, 2020). This recombination rate is a function of co-infection rates, probability of recombination occurring upon co-infection, and selection. As such, the recombination rate we infer here will be (possibly substantially) lower than the within host rate of recombination.”

- The authors also report that recombination rate seems elevated in the Spike protein and suggest that this could be explained by selection favoring recombination events 3' to ORF1ab. This may be true, but the overall discussion on how selection has impacted recombination is quite superficial and seems to ignore some of the recent literature. Tagliamonte et al., for example, have shown how interplay between recombination and negative selection in SARS-CoV evolution have both played a role in increasing covariant dynamic movements of the Spike glycoprotein, while preserving inter-domain structure.

We now expanded the discussion of the literature about recombination in the Introduction and including additional text:

“While the evolutionary purpose of recombination in RNA viruses is not completely understood (Simon-Loriere and Holmes, 2011), there are different explanations of why recombination may be beneficial. In general, recombination is selected for if breaking up the linkage dis-equilibrium is beneficial (Barton, 1995). Recombination can help purge deleterious mutations from the genome, such as proposed by the mutational-deterministic hypothesis (Feldman et al., 1980). It can also increase the rate at which fit combination of mutations occur, such as stated by the Robertson-Hill effect (Hill and Robertson, 1966). Alternatively, recombination in RNA viruses may also just be a by-product of the processivity the viral polymerase (Simon-Loriere and Holmes, 2011).”

We now also added in the Results section that:

“For instance, it has been suggested that selection has acted on multiple recombination events within spike to enhance dynamic molecular movements of the Spike protein (Tagliamonte et al 2021).”

- Rates of adaptation are estimated using mainly the method implemented by Kistler and Bedford (2021) by modifying Bhatt's work. This is a site-counting method that estimates the product between a fixation or polymorphism score and a silent or replacement score, and eventually sums up the products for all sites belonging to a specific frequency class. The method is problematic in the context of the authors' study since fixation and polymorphism scores depend on the number of different nucleotides observed at a site, which in turn likely depends on the sampling size of the population under study. Given the disproportionate amount of SARS-CoV-2 sequence data available, for example, any analysis must select smaller datasets that may or may not be fully representative of the underlying populations, with the unavoidable result of over or under estimating the number of transient polymorphisms, especially when using a method that is not based on phylogeny and cannot distinguish between substitutions occurring on internal (more likely to be fixed) or terminal (more likely to be transient) branches of the tree.

We agree that estimating adaptation this way can be biased for viruses for which there is only a relatively short time since evolution can be tracked. Alongside a limited number recombination events in SARS-CoV-2 thus far we didn't think that there is a lot of information about adaptation and recombination for SARS-CoV-2 at the moment. We now specify more clearly why we didn't perform the same analysis for SARS-CoV-2 and that this way of estimating adaptive evolution requires relatively long sampling intervals for the changes in the consensus genome to represent the long term evolutionary trends. We now write that:

“This approach requires having sequence data gathered over relatively long time periods where the consensus genome allows for an accurate description of the long term evolutionary patterns and, as such, would not be adequate for a pathogen with relatively short evolutionary history, such as for SARS-CoV-2.”

- A paper that proposes a new Bayesian phylogeny method must have a section evaluating performance when analyzing increasingly large sequence data set, computational cost and feasibility, as well exploring potential problems introduced by the often arbitrary choice of priors. None of this is discussed in the present manuscript, which is in my opinion a major flaw. However, I am confident the authors could easily address these points.

We now added a simulation study where we show that the method scales relatively well with the number of samples or the clock rate (which is related to the number of “patterns” in the genome), but scales worse with the number of recombination events, suggesting that, as is, the method is going to be more efficient for larger datasets (more samples) with fewer

recombination events over datasets with many recombination events, such as the long term evolutionary history of SARS-like viruses. This analysis is shown in new Figure S16 and described in the text in Methods under the section "Validation and testing":

"We next tested how the speed of inference scales with the number of recombination events, the number of samples in the dataset and the evolutionary rate. To do so, we simulated 300 recombination networks and sequence alignment of length 10,000 under a Jukes Cantor model with between 10 and 200 leafs and a recombination rate between 1×10^{-5} and 2×10^{-5} recombination events per site per year. This means that for each simulation, there were between 0 and 100 recombination events, allowing us to investigate how the inference scales in different settings. As shown in figure S16, the ESS per hour decreases with the number of recombination events and samples, but not the evolutionary rates. In particular, the ESS per hour decreases much faster with the number of recombination events in a dataset than the number of samples. This suggest that the methods can currently be used more easily to analyze dataset with large number of samples over large number of recombination events."

Additionally, we added a set of simulations and inferences where we compare the inference results between using the correct prior distribution, i.e. the distribution under which the recombination rates for the simulation were sampled, as well as when using the "wrong" prior distributions in new Figure S15. We now write:

"We next tested how the choice of the prior distribution on the recombination rate impacts the recombination rate estimate. To do so, we simulate 20 recombination networks and sequence alignment of length 10000 under a Jukes Cantor model with 100 leafs and a recombination rate drawn randomly from a log-normal distribution. We then infer the recombination rates using 5 different recombination rate priors as shown in figure S15F that put some or a lot of weight on the wrong parameters. As shown in figures S15A-E, we are able to infer recombination rates, even with the wrong priors."

Minor remarks:

- Line 40 "Recombination poses a unique challenge phylogenetic methods" should be corrected into "Recombination poses a unique challenge to phylogenetic methods".

fixed

- Line 86 "A fixed rate of rate of 5×10^{-4} per nucleotide and year" should be replaced by "per nucleotide per year".

fixed

- Line 88 "This means that the sampling times themselves therefore offers, therefore little information" the word "therefore" should be removed.

fixed

- The fixed evolutionary rate chosen by the authors seems reasonable, but I wonder if it would be more appropriate to use a lognormal distribution with confidence intervals matching the uncertainty of current estimates.

We agree that using a range of values would ideal. With the distribution somewhat unknown and there being little to no information from the data about the clock rate itself, the updated estimates would only reflect what we add as uncertainty, not unlike using a fixed clock. In turn, however, the complexity of the inference would increase by quite a bit, which would require substantially longer MCMC runs

- Many Network-based methods, not necessarily based on Bayesian phylogenetics, have been developed and proposed in the literature to investigate and display shared ancestry, such as split decomposition methods. The most relevant should be at least quoted in the introduction to provide a better context to the readers.

We improved the discussion of other methods that deal with recombination in the introduction, including likelihood based and other methods.

Reviewer #3 (Remarks to the Author):

Muller et al present an MCMC method for detecting recombination in virus datasets and apply it to coronaviruses. This approaches infers the underlying reticulate history in the alignment, i.e. explicitly allows for the alternative phylogenetic histories that are taking place. The method is sound and builds an existing theoretical framework. The authors apply their method to several coronavirus data sets, in particular SARS-CoV-2. These results are interesting. Some comments:

Line 23, the Andersen proximal paper addressed whether SCV2 was likely to be natural versus lab-made so not really its origin specifically.

fixed

Line 25 -- Voltz's phylodynamic methods assess virus growth rates not fitness explicitly.

fixed

Line 29 -- mutations are also introduced by anti-viral molecules eg APOBEC, not just replication errors.

Added

Line 78 -- Boni et al. 2020, <https://www.nature.com/articles/s41564-020-0771-4>. show recombination was probably not the origin of SARS-CoV-2 RBM.

Added

We would expect recombination rates to be higher proximal to the regions of the virus genome under selection. Is this what the authors mean when they say these sections have elevated rates? This conflicts with a reporting of a lack of recombination breakpoints in S1 - <https://www.biorxiv.org/content/10.1101/2021.01.22.427830v3>.

We agree that this is in conflict with the inference results from Lytras et al. and now mention this in the results, though we do not really know what would drive these differences. Lytras et al. use a larger dataset including more sarbecoviruses, as well as a very different methodology.

On line 156 the authors write that the evolutionary purpose of recombination is not known but later discuss classical explanations by Barton, Hill & Robertson etc. Please be consistent on this important literature and discuss in the introduction.

We now shifted the discussions about recombination to the introduction and made it more detailed.

The authors should check the SARS-CoV-2 literature for other papers on recombination eg:

<https://www.biorxiv.org/content/10.1101/2020.09.21.300913v2>

<https://www.medrxiv.org/content/10.1101/2021.06.18.21258689v1>

<https://www.biorxiv.org/content/10.1101/2021.01.21.427579v3.full>

<https://www.biorxiv.org/content/10.1101/2020.03.16.993816v2>

<https://www.biorxiv.org/content/10.1101/2020.02.10.942748v3>

We added several references about recombination throughout the manuscript

Given the interest in SARS-CoV-2's ongoing evolution and escape from vaccines, do the authors predict a role for recombination?

Since neither the impact of recombination on vaccine escapes in seasonal coronaviruses, nor the impact of reassortment on escaping flu vaccines is clear, it's complex to have a good prior expectation on what will happen. We now added a small reference in the discussion that "While it's long term impact on the evolutionary dynamics of SARS-CoV-2 remains unclear, the likely

rise of future SARS-CoV-2 recombinants will further necessitate methods that allow to perform phylogenetic and phylodynamic inferences in the presence of recombination”

Reviewers' Comments:

Reviewer #1:

Remarks to the Author:

1. Assuming I am understanding Fig S15 correctly, that five different priors (in panel F) were used to test if the recombination inference recovered the correct simulated recombination parameter, this needs to be highlighted in the main text. I would even put the entire Figure S15 as a main text figure. Authors choice. But this is a pretty important figure.

I would replace "too low", "correct", and "too high" with "low", "centered", and "high".

2. In Figure S4, how many red dots are plotted here? Are these samples from a posterior? Does the inference tell you the posterior for the number of recombination events from 1920 to 2020? Probably. Let's say there are 15 recombination events, and 1000 red dots on the graph. Then one red dot corresponds to something like an 0.015 probability of a recombination event, and 66 red dots clustered together gets you close to the "probability weight" that a recombination event occurred in the place where these red dots are clustered. The reader would like all this information.

Figure S14 is great. And the dark blue densities probably correspond to a probability weight of 1 (or close to it) from the posterior of a recombination event occurring at that time and genome position. If this is true, please say so.

3. Regarding some of the new material on the relationship between recombination and fitness (including comments from Referee 2):

I would not wade into this discussion. Whether recombination speeds up or slows down evolution, or increases or decreases fitness, is strongly dependent on the epistatic effects between loci. There's some recent work by Daniel Weissman (Emory) on this topic (last 5 years, one TPB paper I believe). The Feldman papers on modifier theory are appropriate to cite, as are the Hill-Robertson papers. This even goes back earlier to the Lewontin/Kojima papers of the 1960s. But, none of these papers, to my knowledge, came to a general result and all or almost all acknowledged that the evolution of recombination depends on epistasis. The Barton (1995) paper has a general approach and also seems to acknowledge that epistasis plays an important role (I'm less familiar with this one).

I would simply ignore this entire topic in the paper (as it's not what the main analysis is about) and I would definitely avoid statements like:

"Overall, these results suggest that recombination events are either positively or negatively selected for."

since an event cannot be selected for; only a phenotype can be selected for. A phenotype can be "high propensity for recombination", but this is not tested in the present manuscript.

4. I wouldn't start a sentence with "the evolutionary purpose of .." because we all know that there is no purpose, there is just an outcome of a fitness change (or not) which leads to the fixation (or not) of some trait or genotype. I appreciate that the authors inferred a higher recombination rate in the spike region, but this is not evidence of "whether recombination benefits fitness in human coronaviruses"

Figure 3 is important. This shows that rate of adaptation is positively associated with rate of recombination when scanning across a genome. But, keep the language of 'association' throughout. There is no evidence of causality here.

5. I would not use the word 'rampant' in a sub-heading. Writing could use some improvements in clarity and organization in a few other places. A single editorial sweep at the very end should do it.

Reviewer #3:

Remarks to the Author:

This is a resubmission, the authors have done an excellent job responding to all of the reviewers comments.

REVIEWER COMMENTS

We thank the two reviewers for their positive feedback. We addressed all the remaining concerns of reviewer #1 as detailed below.

Reviewer #1 (Remarks to the Author):

1. Assuming I am understanding Fig S15 correctly, that five different priors (in panel F) were used to test if the recombination inference recovered the correct simulated recombination parameter, this needs to be highlighted in the main text. I would even put the entire Figure S15 as a main text figure. Authors choice. But this is a pretty important figure.

I would replace "too low", "correct", and "too high" with "low", "centered", and "high".

We replace the prior labeling in Figure S15. As suggested, we also moved the figure to the Methods parts in the main text where it's now referenced as Figure 5.

2. In Figure S4, how many red dots are plotted here? Are these samples from a posterior? Does the inference tell you the posterior for the number of recombination events from 1920 to 2020? Probably. Let's say there are 15 recombination events, and 1000 red dots on the graph. Then one red dot corresponds to something like an 0.015 probability of a recombination event, and 66 red dots clustered together gets you close to the "probability weight" that a recombination event occurred in the place where these red dots are clustered. The reader would like all this information.

We now added the suggested addition to the label of Figure S4, adding that: "Each dot represents a probability weight of 0.002% for a recombination event, meaning that 500 dots corresponds to a probability weight of 1 for an event."

Figure S14 is great. And the dark blue densities probably correspond to a probability weight of 1 (or close to it) from the posterior of a recombination event occurring at that time and genome position. If this is true, please say so.

We now changed the coloring of figure S14 and write that: "The contour plots show the density for inferred recombination events for the first 9 iterations of the simulation study. The colors are scaled such that the peak intensity has the same color in all plots.."

3. Regarding some of the new material on the relationship between recombination and fitness (including comments from Referee 2):

I would not wade into this discussion. Whether recombination speeds up or slows down evolution, or increases or decreases fitness, is strongly dependent on the epistatic effects between loci. There's some recent work by Daniel Weissman (Emory) on this topic (last 5 years, one TPB paper I believe). The Feldman papers on modifier theory are appropriate to cite, as are the Hill-Robertson papers. This even goes back earlier to the Lewontin/Kojima papers of the 1960s. But, none of these papers, to my knowledge, came to a general result

and all or almost all acknowledged that the evolution of recombination depends on epistasis. The Barton (1995) paper has a general approach and also seems to acknowledge that epistasis plays an important role (I'm less familiar with this one).

I would simply ignore this entire topic in the paper (as it's not what the main analysis is about) and I would definitely avoid statements like:

"Overall, these results suggest that recombination events are either positively or negatively selected for." since an event cannot be selected for; only a phenotype can be selected for. A phenotype can be "high propensity for recombination", but this is not tested in the present manuscript.

We now distinguish more clearly in the text when we mean recombination as a process is selected for (i.e. why it could have evolved as a process) versus particular recombinant viruses being selected for, i.e. selection increasing frequency of particular recombinant viruses driving increases in frequency of particular recombination events. For example, we now write in the introduction that: "While the reason why the recombination process evolved in RNA viruses is not completely understood, there are different explanations of why recombination may be beneficial." We also clarify in the results section that: "Overall, these results suggest that particular recombination events that have resulted in recombinant viruses are either positively or negatively selected for."

4. I wouldn't start a sentence with "the evolutionary purpose of .." because we all know that there is no purpose, there is just an outcome of a fitness change (or not) which leads to the fixation (or not) of some trait or genotype. I appreciate that the authors inferred a higher recombination rate in the spike region, but this is not evidence of "whether recombination benefits fitness in human coronaviruses"

We deleted the sentence and now start the section writing that: "We next tested whether parts of the genome with higher rates of recombination are also associated with higher rates of adaptation."

Figure 3 is important. This shows that rate of adaptation is positively associated with rate of recombination when scanning across a genome. But, keep the language of 'association' throughout. There is no evidence of causality here.

We changed the section title to "Rates of recombination are associated with rates of adaptation in human seasonal coronaviruses" to make it clearer that this is an association. Additionally, we adapted the text in multiple places say that this is an association. We now write, for example, that "Next, we show that there is an association between higher recombination rates with higher rates of adaptation in human coronaviruses, particularly on the spike protein, suggesting that recombinant viruses are positively or negatively selected based on where breakpoints occur" or "We next tested whether parts of the genome with higher rates of recombination are also associated with higher rates of adaptation"

5. I would not use the word 'rampant' in a sub-heading. Writing could use some improvements in clarity and organization in a few other places. A single editorial sweep at the very end should do it.

We now changed the heading to "Widespread recombination in SARS-like coronaviruses"

Reviewer #3 (Remarks to the Author):

This is a resubmission, the authors have done an excellent job responding to all of the reviewers comments.